# Prevalence and factors associated with exposure to secondhand smoke (SHS) among young people: a cross-sectional study from the Gambia

Isatou K Jallow,[1,2] John Britton,[1] Tessa Langley[1]

[1]UK Centre for Tobacco and Alcohol Studies, Division of Epidemiology and Public Health, University of Nottingham, Nottingham, UK
[2]National Public Health Laboratory, Ministry of Health and Social Welfare (MoH&SW), Banjul, The Gambia

**Correspondence to**
Isatou K Jallow;
jallowisa@hotmail.com

## ABSTRACT

**Background** Annually, 600 000 deaths are attributed to exposure of non-smokers to secondhand smoke (SHS). These include 165 000 among children, about 60% of which occur in Africa and Southeast Asia. As of 2017, only seven countries in the African region had comprehensive smoke-free legislation covering all public places. Given the increasing prevalence of smoking in many low-income countries, preventing exposure to SHS is an urgent public health priority, particularly in Sub-Saharan Africa.

**Objectives** The objective of this study is to obtain a reliable and nationally representative estimate of the prevalence of exposure to SHS and to identify the major risk factors among young people in The Gambia.

**Settings and methods** We used a two-stage cluster random sampling to select students in secondary schools throughout The Gambia and a self-administered questionnaire to collect data on demographic characteristics and detailed indicators of exposure to SHS.

**Results** Of the 10 392 eligible students, 10 289 (99%; 55% girls and 44% boys, age 12–20 years) participated. The proportion of students reporting any exposure to SHS was 97.0% (enclosed public places 59.2%, outdoor public places 61.4%, school 21.3% and home 38.2%), with 96.4% reporting some exposure outside the home. Exposure to SHS in the home was more common in girls and among older students. Parental education, living with parents and being sent to purchase cigarettes were associated with exposure to SHS both within and outside the home. More than 50% of students supported public smoking ban in both enclosed and outdoor public places. About 35% of students were unaware of the harmful effects of exposure to SHS.

**Conclusions** Exposure to SHS is highly prevalent among students in The Gambia and occurs mostly outside of the home. Interventions to reduce SHS exposure in students are urgently needed.

## Strengths and limitations of this study

► This study provides the first comprehensively representative data on the prevalence and determinants of exposure to secondhand smoke among young people in The Gambia.
► The participation rate among those sampled was extremely high, and upper basic schools and senior secondary schools were sampled from schools throughout the country.
► Self-administered questionnaires were used; students may have under-reported or over-reported their answers.
► The survey was limited to students; it may not represent the smoking prevalence of all young people in this age group.

Asia.[2] SHS is associated with diseases such as respiratory and cardiovascular disease, lung cancer and other forms of cancers and accounts for about 1% of the global burden of disease.[3]

In 2004, 40% of children, 33% male and 35% female non-smokers were exposed to indoor secondhand tobacco smoke worldwide.[1] The WHO Framework Convention on Tobacco Control has established that 100% smoke-free environments are the only proven way to adequately protect people from the harmful effects of secondhand tobacco smoke.[4] Despite the progress made in smoke-free policy adoption, the populations of three quarters of all countries, including 88% of low-income countries, are vulnerable to the dangers of SHS due to weak or absent smoke-free laws.[5] Currently, only seven countries in the African region has comprehensive smoke-free legislation covering all types of public places or at least 90% of the population covered by complete subnational smoke-free legislations.[6] Given the increasing prevalence of smoking in many low-income countries, preventing exposure to SHS is an

## BACKGROUND

Annually, six million people die from tobacco use. Of these, an estimated 600 000 deaths are attributed to exposure of non-smokers to secondhand smoke (SHS),[1] including 165 000 deaths among children, of which about 60% occur in Africa and Southeast

urgent public health priority in these countries, particularly in Sub-Saharan Africa.

The Gambia is a West African country of 1.9 million people with a per capita gross domestic product of US$471 in 2015.[7] Since 1999, the Gambia has been implementing the Prohibition of Smoking (Public Places) Act 1998, which prohibits tobacco smoking in public places, workplaces, hospitals, public vehicles and in government properties or premises.[8] However, data on the prevalence of exposure to secondhand tobacco smoke and the determinants of SHS are limited. In the Global Youth Tobacco Survey (GYTS), which was conducted in the Greater Banjul Area in 2008, 45% of students stated that people smoked in their presence at home and 59% were exposed to other people's smoke outside their homes.[9] However, these estimates are out of date and the authors are not aware of any studies about the determinants of exposure to SHS among adolescents in The Gambia. To obtain a reliable and nationally representative estimate of the prevalence of exposure to SHS and to identify the major risk factors among young people in The Gambia, we conducted a survey of SHS prevalence and determinants in a nationwide sample of Gambian schools.

## METHODS
### Study population
This study was carried out in a sample of upper basic schools (UBS) and senior secondary schools (SSS) throughout The Gambia: Banjul and Kanifing municipalities and the rest of the country, which comprise five regions using methods described previously.[10] In brief, a nationally representative sample of students in UBS (grades 7–9) and SSS (grades 10–12), aged 12 to 20 years were generated by a two-stage cluster sampling. In the first stage, a list of all UBS and SSS was obtain from the Ministry of Basic and Secondary Education, and schools were randomly selected from the list of schools with a probability proportional to their enrolment size. In the second stage, classes within the selected schools were randomly selected from the total number of classes in the schools. All students in the selected classes that attended school on the day of the survey were eligible to participate. Our study was powered to estimate youth smoking prevalence of 15% with 1% precision, which required a minimum sample size of 4885 (Epi Info V.7).

### Data collection and study variables
Participating students completed a self-administered questionnaire adapted from the GYTS, collecting data on a range of variables including demographic details, exposure to SHS, support for public smoking regulations and knowledge of the harmful effect of SHS. The questionnaire also included series of questions on several indicators of tobacco use, smoking susceptibility, exposure to tobacco advertisements and promotion, antismoking media messages, beliefs about the danger of smoking and

the perceived benefits of smoking; these data have been reported in a separate publication.[10]

Self-reported exposure to SHS was the outcome variable and was assessed in the study by the following questions: "During the past 7 days, on how many days has anyone smoked in your presence?: inside your home, in an outdoor public place, in an indoor public place, inside any public transportation'; and 'during the past 30 days has anyone smoked in your presence inside the school buildings or premises?". Exposure to SHS was defined as being exposed to SHS on at least 1 day in the past 7 days in any public place and in the home or in the past 30 days at school. Exposure to SHS outside the home was defined as any exposure at outdoor and indoor public places, inside any public transportation and at school. The independent variables used in the study were gender, age and religion, school level, school funding type, school locality, parents' educational level, tobacco use by family and friends, sent to purchase cigarettes and support for smoke-free bans.

The survey was carried out between June and December 2016. The survey was approved by The Gambia Government/Medical Research Council Joint Ethics Committee and by the Ethics Committee of the Faculty of School of Medicine and Health Sciences of the University of Nottingham, UK.

### Statistical analysis
Data were analysed in Stata V.14. Proportions and 95% CIs were obtained as estimates of the prevalence of exposure to SHS. We adjusted associations for a priori confounders comprising age, gender and rural/urban area of schools and used multivariate logistic regression analyses to predict factors associated with exposure to SHS.

## RESULTS
### Characteristics of the study population and prevalence of exposure to SHS
A total of 50 schools throughout the country participated in the study, including 33 UBS and 17 SSS, comprising 13 private, 27 public and 10 grant-aided schools. A total of 10 395 students were registered in the selected classes, of which 10 289 (99%) students participated in the study. Detailed characteristics of the study participants are summarised in table 1. Among the total sample, 55.6% were girls and 44.4% were boys. More than half (63.9%) of participants were aged between 14 and 17 years. The majority (74.6%) of the students attended public schools were of Muslim faith (93.1%), lived with their parents (80.2%), lived in homes where smoking was not allowed (70.9%) and had no family members (71.6%) or friends (66.5%) who smoked. About half (43.4%) of the students reported purchasing cigarettes for their parents or others and 97.0% of students were exposed to SHS. About 35% of students had seen people smoking inside their school and about a third of those who had seen people smoking

**Table 1** Sociodemographic characteristics of study participants and SHS exposure (n=10 289)

| Characteristics | Categories | Total (n=10289) | % |
|---|---|---|---|
| Age group | 12 –13 | 960 | 9.3 |
| | 14–15 | 2776 | 6.9 |
| | 16–17 | 3812 | 37.0 |
| | 18–19 | 2221 | 21.5 |
| | 20 | 525 | 5.1 |
| School type | UBS | 5785 | 56.2 |
| | SSS | 4504 | 43.7 |
| School funding | Public | 7678 | 74.6 |
| | Grant aided | 1052 | 10.5 |
| | Private | 1559 | 15.1 |
| School locality | Rural | 2453 | 23.8 |
| | Urban | 7833 | 76.1 |
| Religion | Muslim | 9564 | 93.1 |
| | Christian | 602 | 5.8 |
| | Other | 103 | 1.0 |
| Living with parents | Yes | 8250 | 80.2 |
| | No | 2029 | 19.7 |
| Father's education | No formal education | 2420 | 23.5 |
| | Primary school | 674 | 6.5 |
| | Secondary school | 2120 | 20.6 |
| | Tertiary | 1867 | 18.1 |
| | Quranic/Arabic school | 2097 | 20.3 |
| | Do not know | 1110 | 10.7 |
| Mother's education | No formal education | 3022 | 29.3 |
| | Primary school | 1168 | 11.3 |
| | Secondary school | 2220 | 21.5 |
| | Tertiary | 1024 | 9.9 |
| | Quranic/Arabic school | 1772 | 17.2 |
| | Do not know | 1080 | 10.5 |
| Smoking status | Non-smokers | 8565 | 83.2 |
| | Ever smokers | 1719 | 16.7 |
| Home smoking rules | No | 7295 | 70.9 |
| | Sometimes | 1085 | 10.5 |
| | Yes | 1906 | 8.5 |
| Family smoking | None | 7364 | 71.6 |
| | Mother | 274 | 2.6 |
| | Father | 1199 | 11.6 |
| | Brother/Sister | 718 | 6.9 |
| | Others | 729 | 7.0 |

Continued

**Table 1** Continued

| Characteristics | Categories | Total (n=10289) | % |
|---|---|---|---|
| Number friends who smoke | None | 6790 | 66.0 |
| | One | 673 | 6.5 |
| | Two | 356 | 3.4 |
| | Three or more | 762 | 7.4 |
| | Not sure | 1699 | 16.5 |
| Sent to buy cigarettes for parents or others | No | 5816 | 56.6 |
| | Yes | 4459 | 43.4 |
| Exposure to SHS | Exposed | 9982 | 97.0 |
| | Not exposed | 304 | 3.0 |
| Seen anyone smoking at school | Yes | 3604 | 35.0 |
| | No | 6666 | 64.9 |
| Person seen smoking at school | Friends | 807 | 7.8 |
| | Other students | 891 | 8.6 |
| | Teachers | 1232 | 12.0 |
| | Other staff | 674 | 6.5 |
| | None | 6666 | 64.9 |

SHS, secondhand smoke; SSS, senior secondary schools; UBS, upper basic schools.

in their schools (12.1% of the total sample) had seen teachers smoking.

### Participants' place of exposure to SHS

Figure 1 describes the participants' exposure to SHS in different locations. More than half of the students were exposed to SHS for at least 1 day in the past week in enclosed (59.2%) and outdoor (61.4%) public places and 38.2% in the home. About 96.4% of students were exposed to SHS outside the home (enclosed and outdoor public places, public transportation and school) and 95.7% of the students were exposed to SHS in an enclosed place (enclosed public place, public transportation, school buildings and/or at home). About 1 in 5 (21.3%) students had been exposed to SHS at school on at least 1 day in the previous 30 days.

### Frequency of exposure to SHS among study participants

The frequency of SHS exposure at home and school is summarised in table 2. Approximately 8% of students reported their father smoking in their presence, and 4.7% of students reported their mother smoking in their presence, every day in the past 7 days. Daily exposure to SHS from other family members (11.7%) was much higher compared with exposure from parents. About 4.4% of students were exposed to SHS every day in school buildings or premises.

Participants' perceptions of the risk of exposure to SHS and support for ban on public smoking are also outlined in table 2. One in 4 (26.6%) and 1 in 10 (9.4%) participants reported that exposure to SHS was definitely not harmful and probably not harmful,

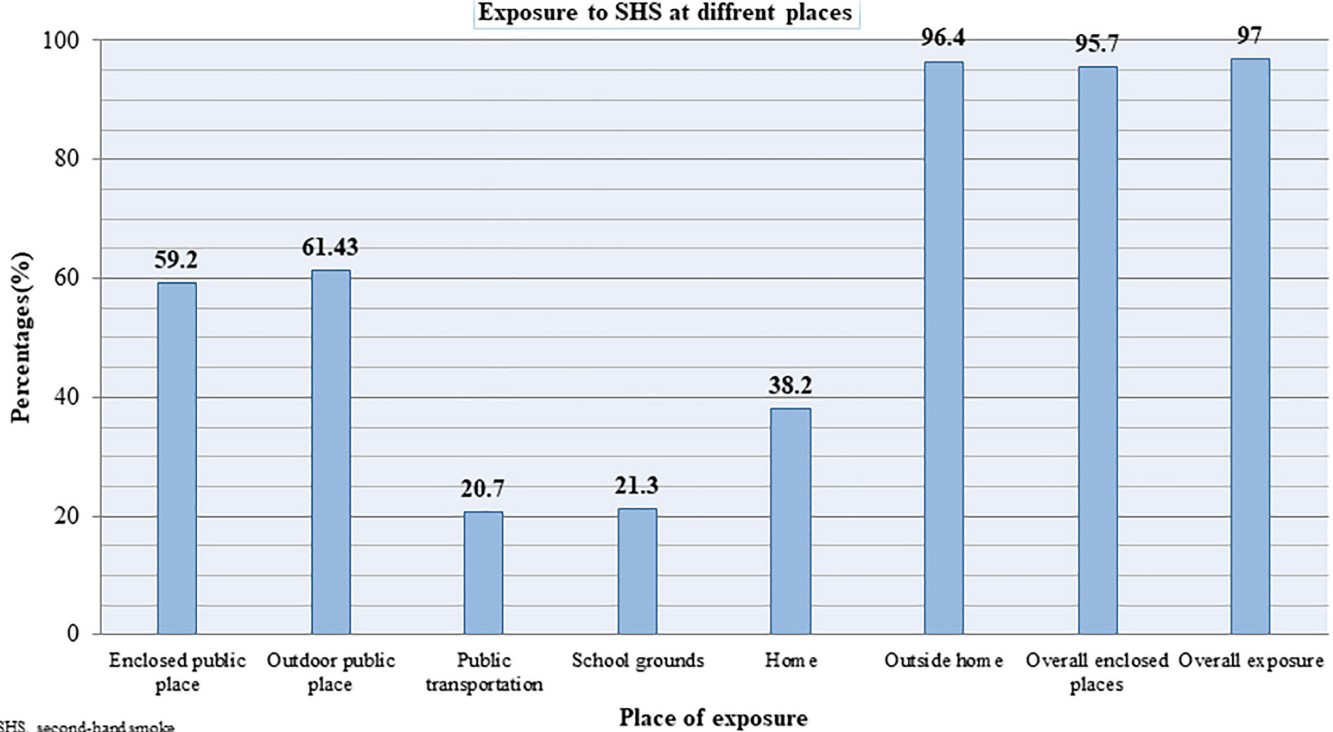

**Figure 1** Exposure to SHS at different locations. SHS, secondhand smoke.

respectively. About half of the participants supported a smoking ban in enclosed (56.0%) and outdoor (56.9%) public places.

### Factors associated with SHS exposure at home

As shown in table 3, after adjusting for age, gender and school location, girls (OR 1.34, 95% CI 1.22 to 1.47), students aged 18–20 years (OR 1.20, 95% CI 1.02 to 1.40), those in UBS schools (OR 1.40, 95% CI 1.25 to 1.57) and student attending grant-aided schools (OR 1.36, 95% CI 1.17 to 1.58) were significantly more likely to be exposed to SHS. Living with parents (OR 0.83, 95% CI 0.74 to 0.93), being a smoker (OR 1.63, 95% CI 1.31 to 2.03), having smoking allowed at home and having family members or friends who smoked also significantly increased the risk of students exposure to SHS at home. In addition, students who were sent to purchase cigarettes (OR 1.98, 95% CI 1.80 to 2.18) and supported a ban on public smoking were significantly more likely to be exposed to SHS at home.

### Factors associated with SHS exposure outside the home

Outside the home, lower maternal and higher paternal educational level, living with parents and being sent to purchase cigarettes for others (OR 1.42, 95% CI 1.14 to 1.77) were significantly associated with increased risk of exposure to SHS (table 3). In addition, older students aged 18–20 (OR 1.14, 95% CI 0.83 to 1.56) were more likely to be exposed to SHS outside the home compared with younger students aged 12–14.

### DISCUSSION

This is the first study to provide detailed data on exposure to SHS in a nationally representative sample of adolescent school students in The Gambia. We found a very high level of self-reported exposure to SHS among students, and, contrary to expectation, found that while around two in five respondents reported SHS exposure in the home, a large majority of young people reported exposure in public places. Older students in our sample were generally more likely to be exposed to SHS, as were children under the age of 15. Older students and girls were significantly more like to be exposed to SHS at home compared with boys. Students in our sample were also more likely to be exposed to SHS if their family members or friends smoked, if they attended UBS or grant-aided schools, smoking was allowed in the home and among those who were not Muslim. Exposure to SHS at home and outside the home were also associated with parental educational level, though in opposite ways; higher maternal and lowest paternal levels of education were associated with lower exposure. Students who were sent to purchase cigarettes for others were also more likely to be exposed to SHS. Awareness of the harm to health of SHS exposure was low; with more than a quarter of students reporting that exposure was probably or definitely not harmful. However, most students supported a smoking ban in both enclosed and outdoor public places.

Our study has some limitations. This was a cross-sectional study and we used a self-administered questionnaire

**Table 2** Frequency of exposure to SHS among participants and support for public smoking ban (n=10 289)

| Characteristics | Total (n=10 289) N | (%) |
|---|---|---|
| **Exposure to SHS at home** | | |
| Father | | |
| About every day | 800 | 7.7 |
| Sometimes | 1253 | 12.1 |
| Never | 6265 | 60.9 |
| Do not have/do not see this person | 1958 | 19.0 |
| Mother | | |
| About every day | 489 | 4.7 |
| Sometimes | 951 | 9.2 |
| Never | 6834 | 66.5 |
| Do not have/do not see this person | 2000 | 19.4 |
| Sibling | | |
| About every day | 468 | 4.5 |
| Sometimes | 1060 | 10.3 |
| Never | 6848 | 66.6 |
| Do not have/do not see this person | 1901 | 18.5 |
| Others | | |
| About every day | 1210 | 11.7 |
| Sometimes | 3223 | 31.3 |
| Never | 3802 | 37.0 |
| Do not have/do not see this person | 2309 | 19.8 |
| **Exposure to SHS at school** | | |
| Inside school buildings | | |
| About every day | 516 | 5.0 |
| Sometimes | 1905 | 18.5 |
| Never | 5339 | 52.0 |
| Do not know | 2506 | 24.1 |
| School premises | | |
| About every day | 459 | 4.4 |
| Sometimes | 1839 | 17.8 |
| Never | 5218 | 50.7 |
| Do not know | 2762 | 26.8 |
| **Support for public smoking band and perception of risk of exposure to SHS** | | |
| Thinks smoking should be banned in enclosed public places | | |
| Yes | 5761 | 56.0 |
| No | 4517 | 43.9 |
| Thinks smoking should be banned in outdoor public places | | |

Continued

**Table 2** Continued

| Characteristics | Total (n=10 289) N | (%) |
|---|---|---|
| Yes | 5852 | 56.9 |
| No | 4424 | 43.0 |
| Thinks SHS is harmful | | |
| Definitely not | 2736 | 26.6 |
| Probably not | 968 | 9.4 |
| Probably yes | 1296 | 12.6 |
| Definitely yes | 5278 | 51.3 |

SHS, secondhand smoke.

to measure exposure to SHS: students may have under or over reported the answers. However, students self-reports of exposure to SHS has been reported to be highly consistent with urinary cotinine level measurement.[11] Our sampling method ensured that the population selected was likely to be highly representative of young people in The Gambia while we recognised that this limits the generality of our findings to young people not in school. Data from the Ministry of Basic and Secondary Education indicated gross enrolment rates of 68.1% and 41.2% for UBS and SSS, respectively, and the universal primary and secondary education initiative which have seen greater number of young people go to school in The Gambia.[12] Furthermore, the present study has a number of strengths that include a large sample size and high response rate among those interviewed, which also supports the robustness of the study findings. Additionally, the study addressed SHS exposure both at home and outside the household.

Previous studies of smoking and exposure to SHS among students in The Gambia are limited, the most recent and widely quoted being the 2008 GYTS survey. The current study estimated a lower overall prevalence of exposure to SHS than ours, but this could well reflect the restricted local sampling frame used in the GYTS.[9] The high prevalence of exposure to SHS is consistent with earlier studies in The Gambia and other countries in Africa.[13 14]

In The Gambia, the Public Smoking Act, which bans smoking in all public places, came into effect in 1998. However, our observation that exposure to SHS remains high, and may even have increased since the 2008 GYTS, suggests that efforts are still needed to enhance the enforcement of this law, particularly since public places were the most frequent source of exposure to SHS among young people in The Gambia. Beyond the direct health benefit of smoke-free policies, implementing smoke-free laws, especially in public places, has been shown to change the public acceptance of smoking by the general population.[15 16] Most countries in the African region still have weak or even non-existent smoke-free laws, and compliance with smoke–free laws varies extensively.[6]

**Table 3** Determinants of SHS exposure at home and outside the home among participants (n=10 289)

| Characteristics | Categories | Total (n=9982) | Home | | Outside home | |
|---|---|---|---|---|---|---|
| | | | Adjusted OR (95% CI) | P value | Adjusted OR (95% CI) | P value |
| Age group | 12–14 | 2184 (96.8) | 1 | 0.010 | 1 | <0.001 |
| | 15–17 | 5129 (97.0) | 1.07 (0.95 to 1.21) | | 1.04 (0.80 to 1.35) | |
| | 18–20 | 2669 (97.1) | 1.20 (1.02 to 1.40) | | 1.14 (0.83 to 1.56) | |
| Gender | Boys | 4437 (97.1) | 1 | <0.001 | 1 | 0.890 |
| | Girls | 5545 (96.1) | 1.34 (1.22 to 1.47) | | 1.01 (0.81 to 1.26) | |
| School type | SSS | 4380 (97.2) | 1 | <0.001 | 1 | 0.119 |
| | UBS | 5602 (96.8) | 1.40 (1.25 to 1.57) | | 0.80 (0.61 to 1.05) | |
| School funding | Public | 7464 (97.2) | 1 | <0.001 | 1 | 0.121 |
| | Grant aided | 991 (94.2) | 1.36 (1.17 to 1.58) | | 0.59 (0.43 to 0.80) | |
| | Private | 1527 (98.1) | 0.80 (0.70 to 0.92) | | 1.47 (1.00 to 2.17) | |
| School locality | Rural | 2389 (97.3) | 1 | <0.001 | 1 | 0.360 |
| | Urban | 7593 (96.9) | 0.71 (0.64 to 0.79) | | 0.88 (0.67 to 1.15) | |
| Religion | Muslim | 9277 (96.9) | 1 | 0.049 | 1 | 0.346 |
| | Christian | 588 (97.6) | 1.22 (1.01 to 1.47) | | 1.16 (0.69 to 1.95) | |
| | Other | 98 (95.1) | 1.35 (0.86 to 2.12) | | 0.56 (0.24 to 1.32) | |
| Father's education | No education | 2326 (96.1) | 1 | 0.150 | 1 | <0.001 |
| | Primary | 655 (97.1) | 0.95 (0.78 to 1.16) | | 2.01 (1.24 to 3.26) | |
| | Secondary | 2063 (97.3) | 1.26 (1.09 to 1.45) | | 2.36 (1.69 to 3.31) | |
| | Tertiary | 1827 (97.8) | 0.88 (0.75 to 1.04) | | 3.46 (2.29 to 5.22) | |
| | Quranic/Arabic | 2033 (96.9) | 0.98 (0.85 to 1.14) | | 2.49 (1.77 to 3.50) | |
| | Do not know | 1077 (97.0) | 1.14 (0.95 to 1.36) | | 2.12 (1.39 to 3.23) | |
| Mother's education | No education | 2941 (97.3) | 1 | <0.001 | 1 | <0.001 |
| | Primary | 1137 (97.3) | 1.38 (1.18 to 1.61) | | 0.94 (0.62 to 1.43) | |
| | Secondary | 2176 (98.0) | 0.92 (0.81 to 1.04) | | 0.75 (0.52 to 1.08) | |
| | Tertiary | 1009 (98.5) | 0.91 (0.77 to 1.07) | | 0.41 (0.26 to 0.65) | |
| | Quranic/Arabic | 1676 (94.5) | 0.95 (0.83 to 1.08) | | 0.30 (0.21 to 0.42) | |
| | Do not know | 1040 (96.2) | 0.83 (0.71 to 0.98) | | 0.45 (0.30 to 0.69) | |
| Living with parents | No | 7988 (96.8) | 1 | 0.001 | 1 | 0.008 |
| | Yes | 1987 (97.9) | 0.83 (0.74 to 0.93) | | 0.66 (0.49 to 0.89) | |
| Smoking status | Non-smokers | 8309 (97.0) | 1 | <0.001 | 1 | 0.250 |
| | Ever smokers | 1671 (97.2) | 1.63 (1.31 to 2.03) | | 1.46 (0.76 to 2.82) | |
| Smoking at home allowed | No | 7060 (96.7) | 1 | <0.001 | 1 | 0.524 |
| | Sometimes | 1070 (98.6) | 2.29 (1.99 to 2.64) | | 0.88 (0.62 to 1.25) | |
| | Yes | 1849 (97.0) | 2.73 (2.43 to 3.07) | | 0.86 (0.65 to 1.14) | |
| Family smoking | None | 7112 (96.5) | 1 | <0.001 | 1 | 0.060 |
| | Mother | 270 (98.5) | 1.66 (1.27 to 2.18) | | 2.64 (0.95 to 7.29) | |
| | Father | 1177 (98.1) | 3.16 (2.74 to 3.65) | | 0.75 (0.54 to 1.04) | |
| | Sibling | 702 (97.7) | 1.76 (1.49 to 2.08) | | 1.14 (0.72 to 1.81) | |
| | Others | 716 (98.2) | 2.09 (1.77 to 2.47) | | 1.33 (0.80 to 2.21) | |
| Number of friends who smoke | None | 6563 (96.6) | 1 | <0.001 | 1 | 0.062 |
| | One | 653 (97.0) | 1.89 (1.58 to 2.26) | | 1.11 (0.71 to 1.73) | |
| | Two | 344 (96.6) | 1.94 (1.52 to 2.47) | | 1.14 (0.63 to 2.08) | |
| | Three or more | 743 (97.5) | 1.53 (1.28 to 1.82) | | 1.39 (0.85 to 2.25) | |
| | Not sure | 1670 (98.2) | 1.13 (1.00 to 1.28) | | 1.63 (1.15 to 2.30) | |

Continued

**Table 3** Continued

| Characteristics | Categories | Total (n=9982) | Home Adjusted OR (95% CI) | P value | Outside home Adjusted OR (95% CI) | P value |
|---|---|---|---|---|---|---|
| Sent to buy cigarettes | No | 5700 (98.0) | 1 | <0.001 | 1 | 0.001 |
| | Yes | 4271 (95.7) | 1.98 (1.80 to 2.18) | | 1.42 (1.14 to 1.77) | |
| Ban at enclosed public places | No | 5638 (97.8) | 1 | <0.001 | 1 | 0.191 |
| | Yes | 4336 (95.9) | 1.22 (1.09 to 1.37) | | 1.19 (0.91 to 1.55) | |
| Ban at outdoor public places | No | 5713 (97.6) | 1 | 0.024 | 1 | 0.760 |
| | Yes | 4259 (96.2) | 0.88 (0.78 to 0.98) | | 1.04 (0.80 to 1.35) | |

SHS, secondhand smoke; SSS, senior secondary schools; UBS, upper basic schools.

Furthermore, enforcement of smoke-free polices in most African countries have been identified as a major challenge.[17]

Similar to previous findings,[18] our study also showed that more than half of the students are supportive of polices that ban public smoking; however, many are unaware of the harmful effects of exposure to SHS. Adolescence awareness of the harmful health effects of SHS has been shown to be associated with a reduced risk of exposure to SHS[19 20] and suggests that improved education on the risks of SHS could lead to reductions in exposure.

We found that older students were more likely to be exposed to SHS both outside the home and inside the home; this is consistent with findings in previous studies among students.[21 22] Older students have more opportunities to be outside the home in public places where smoking is more likely to happen. Our finding that parents' educational level, living with parents and being sent to purchase cigarette for others were significant determinants of exposure to SHS in public places is consistent with previous studies[18 23 24] and probably arises from the fact that these characteristics all identify contact with others who smoke.

All of the participant schools in this study reported that they had implemented a comprehensive smoke–free campus policy, yet more than a quarter of students reported SHS exposure at school. These findings, which are consistent with previous reports of significant exposure to SHS at school,[25–27] suggest that enforcement of school-based tobacco control measures needs to be strengthened. Studies have shown that in schools with comprehensive policies and high compliance, students are much less likely to report exposure and report lower intentions to smoke in the future.[28]

Our results showed that parents' educational level, family or friends' smoking status, living with parents, home smoking rules and being sent to purchase cigarette for others were significant determinants of exposure to SHS in the home and is consistent with previous studies.[18 21–24] Furthermore, it has also been shown that non-smokers exposed to SHS at home are more likely to be susceptible to initiating smoking than those not exposed.[29] Educating parents about the harmful effects of smoking and exposure to SHS could be one of the effective ways to protect young people at home. This will help to protect children, help parents who smoke to quit and discourage others from smoking in their homes.

This study has shown that exposure to SHS is very high among students and that despite smoke-free laws, protection against SHS exposure in public places in The Gambia is still inadequate. There is an urgent need to advocate for interventions to reduce the current level of exposure to SHS and minimise further exposure among students. This underscores the need to develop comprehensive smoke-free laws and strictly enforce these laws in all environments. Further research is required to determine whether this is a problem among students alone or reflects a wider pattern of exposure to SHS among the general population.

**Acknowledgements** We thank all the students who participated and the teachers for helping in the coordination and also all the regional education directors and the Ministry of Basic and Secondary Education. We profoundly recognise the contributions of the field workers and staff of the National Public Health Laboratory (NPHL). We also acknowledged the support of the Tobacco Control Unit of the Ministry of Health and Social Welfare.

**Contributors** IKJ: study design, data collection, analysis and drafting of manuscript; TL and JB: study design and revision of manuscript.

**Funding** This research received no specific grant from any funding agency in the public, commercial or not-for-profit sectors.

**Competing interests** None declared.

**Patient consent** Not required.

**Ethics approval** The Gambia Government/Medical Research Council (MRC) Joint Ethics Committee.

**Provenance and peer review** Not commissioned; externally peer reviewed.

**Data sharing statement** No additional data are available.

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
