## [Reviewer comments · BMJ Open]

ARTICLE DETAILS

TITLE (PROVISIONAL)	Prevalence and factors associated with exposure to Second-Hand Smoke (SHS) among young people: a cross-sectional study from The Gambia
AUTHORS	Jallow, Isatou; Britton, John; Langley, Tessa

VERSION 1 – REVIEW

REVIEWER	Ellis Owusu-Dabo Kwame Nkrumah University of Science and Technology, Kumasi, Ghana
REVIEW RETURNED	21-Sep-2017

GENERAL COMMENTS	Very interesting paper showing novelty in the approach and findings. However, Authors should discuss the sampling process in detail and as well should also discuss the lack of 'objectivity' in the assessment of exposure to SHS. There are several methods of measuring objectively exposure to SHS. Was this done? There was no means of verification other than from the self administered questionnaire. There should be a discussion on this and limitations of this further elucidated in the write-up.
--

REVIEWER	Hadii Mamudu East Tennessee State University, USA
REVIEW RETURNED	28-Nov-2017

GENERAL COMMENTS	Prevalence and determinants of exposure to second-hand smoke (SHS) among young people: a cross-sectional study from The Gambia Brief Summary This study utilizes a survey data of 10,289 students to examine exposure to SHS inside and outside home and tease out factors associated with such exposure. It was found that 97.0% has been exposed to any form of SHS, with 96.4% reporting exposure outside the home. While there were differences in factors associated with exposure at home or outside home, parental education, living with parents, and been sent to purchase cigarette were significantly associated with increased odds of exposure both inside and outside home. The following are my comments: General comment SHS is an established health hazard and there is no known safe level of exposure, which means that it is important to investigate how people are exposed to it.
---

Thus, the strengths of this paper include:

1. The paper is well-written with good logical flow.
2. The study involved a country where the last local study on tobacco use and SHS exposure was in 2008 (GYTS). In this respect, there is scarcity of evidence to inform policy and advocacy, making the paper important for the country.
3. Unlike the 2008 study, this current study utilizes a large nationwide representative sample of school-going children, which suggests that the study can be generalized to students in the country.
4. The outcomes (SHS exposure inside or outside the home) is important and the analytical approach is appropriate.

In spite of the aforementioned strengths, the study has certain weaknesses:

1. The results do not provide any novel evidence about SHS exposure, although it could be of significance in the country and results on the frequency of SHS exposure is interesting.
2. This is a cross-sectional study with several limitations; however, the authors did not discuss them.
3. The study involved only school-going children, which limits generalization to include those who do not attend school.
4. There is lack of information on how the variables were ascertained, measured, recoded etc. Especially, there is no information on the validity and reliability of the measures the authors used for the study. If these measures have already been validated, please provide such information in the paper.

Specific comments

Abstract

1. Ln 37: "21..3%" should be "21.3%"

Introduction

1. Page 4 of 19, para 2: The authors utilize 2011 information for the countries in Africa with comprehensive smoke-free legislation. This is too old and they should use up-to-date information (see Lns 30-34 on page 14 of 19 that utilizes different information).

Methods

1. How were the independent variables selected? More information on that is needed.
2. You have indicated in the statistical analysis section that "... multivariate logistic regression analyses to ascertain the predicting factors of SHS exposure". Again, this is a cross-sectional, not longitudinal study. As such, the study can only establish association and nothing more.

Results

1. Is there any scientific/biological reason for the classification of age in the study? If not, it will be better to utilize the classification used in GYTS studies to facilitate comparison and ensure future meta-analysis.
2. Please revise the entire section on "Factors associated with exposure to SHS" to ensure that it reflects an associational study. Also, in this revision, you should not only indicate whether the relationship is significant or not but also you should include the direction of the association. E.g., X is significantly associated with increased/decreased Y.

Discussion

1. The reference #17 on page 14 of 19 is likely wrong. Please look for correct and more appropriate reference.
2. Page 15 and 19, para 2: The authors should reconsider the suggestion of having government to develop policies for private homes as it could raise civil liberty and social issues.

	References Although I will defer to the authors on this, references 17 and 18 utilize information from Saudi Arabia study, which may not be correct. As such, I will urge the authors to ensure that the references that they have used correctly support the information provided. Minor comments  1. The usage of “determinants” in the title suggests some form of causation. The study by the authors can establish only some form of association. As such, the authors should revise the title to reflect the kind of study that they conducted. 2. Ensure consistency in the spelling of second-hand smoke (see page 8 of 19, ln 49). 3. Page 11 of 19, ln 34: Provide space between “faiths” and “(OR....)” 4. Ensure consistency in the use of decimal places (see page 13 of 19, ln 52) 5. Page 15 of 19: “effect” on ln 42 should be “effective”. 6. Spell out all abbreviations in the tables and figure or include a note. 7. Add “(N=10,289) in the title of all the tables.
--	---

VERSION 1 – AUTHOR RESPONSE

Reviewer: 1

However, Authors should discuss the sampling process in detail and as well should also discuss the lack of 'objectivity' in the assessment of exposure to SHS. There are several methods of measuring objectively exposure to SHS. Was this done? There was no means of verification other than from the self-administered questionnaire. There should be a discussion on this and limitations of this further elucidated in the write-up.

More details on the sampling process and the SHS measures used in the study have been added to the method section as outlined above. Due to funding and logistics we were not able to use objective methods to measure exposure to SHS. However the method we used to measure SHS in our study was adapted from the Global Youth Tobacco Survey which has been shown to have high validity and reliability. This limitation has been acknowledged in the added paragraph on the limitations of the study (see below). Furthermore the present study has a number of strengths that include a relatively large sample size and high response rate among those interviewed, which also supports the robustness of the study findings. In addition, the study addressed SHS exposure both at home and outside the household.

Reviewer: 2

In spite of the aforementioned strengths, the study has certain weaknesses:

1. The results do not provide any novel evidence about SHS exposure, although it could be of significance in the country and results on the frequency of SHS exposure is interesting.

Thank you for your comment and we do acknowledge your concern. This is the first study that has measured the frequency of exposure to SHS in the Gambia and has provided the baseline data for policy makers and future studies.

2. This is a cross-sectional study with several limitation; however, the authors did not discuss them. A paragraph on the limitations of the study has been added and it reads as follows:

Our study has some limitations. This was a cross-sectional study and we used a self-administered questionnaire to measure exposure to SHS: students may have under- or over-reported the answers. However students' self-report of exposure to SHS has been reported to be highly consistent with urinary cotinine level measurement.

(11) Our sampling method ensured that the population selected was likely to be highly representative of young people in The Gambia. While we recognise that this limits the generalizability of our findings to young people not attending school, data from the Ministry of Basic and Secondary Education (MoBSE) indicate gross enrolment rates of 68.1% and 41.2% for UBS and SSS respectively and the universal primary and secondary education initiative which has seen greater number of young people go to school in the Gambia. (12) Furthermore the present study has a number of strengths that include a large sample size and high response rate among those interviewed, which also supports the robustness of the study findings. In addition, the study addressed SHS exposure both at home and outside household.

3. The study involved only school-going children, which limits generalization to include those who do not attend school.

This limitation has been acknowledged and is included in the paragraph above.

4. There is lack of information on how the variables were ascertained, measured, recoded etc. Especially, there is no information on the validity and reliability of the measures the authors used for the study. If these measures have already been validated, please provide such information in the paper.

A detailed section of how the outcome variable was measured has been added in the method section and references have been added that indicated the validity and reliability of the measures we used.

Specific comments

Abstract

1. Ln 37: "21..3%" should be "21.3%" The suggested correction has been made.

Introduction

1. Page 4 of 19, para 2: The authors utilize 2011 information for the countries in Africa with comprehensive smoke-free legislation. This is too old and they should use up-to-date information (see lns 30-34 on page 14 of 19 that utilizes different information).

The suggested correction has been made. Up-to-date information with a new reference has been inserted and this sentence now reads as follows: "Currently only seven countries in the African region have comprehensive smoke-free legislation covering all types of public places or at least 90% of the population covered by complete sub-national smoke free legislations".

Methods

1. How was the independent variables selected? More information on that is needed.

The independent variables used in the study were socio-demographic characteristics; gender, age and religion, school level, school funding type, school locality, parents educational level, tobacco use by family and friends, sent to purchase cigarettes and support for smoke-free bans. The variables were adapted from the Global Youth Tobacco Survey and similar previous studies.

2. You have indicated in the statistical analysis section that "... multivariate logistic regression analyses to ascertain the predicting factors of SHS exposure". Again, this is a cross-sectional, not longitudinal study. As such, the study can only establish association and nothing more.

This sentence has been revised and now reads: We adjusted associations for a priori confounders comprising age, gender and rural/urban area of school, and used multivariate logistic regression analyses to predict factors associated with exposure to SHS.

Results

1. Is there any scientific/biological reason for the classification of age in the study? If not, it will be better to utilize the classification used in GYTS studies to facilitate comparison and ensure future meta-analysis.

Thank you for raising this concern. The study adapted the GYTS methodology which focuses on adolescent of ages 13-15 and corresponding grades. In the Gambia this corresponds to Upper Basic and Senior Secondary schools covering grades 7 -12.

Also in the Gambia Children enter grade 1 at the age of 7 and annual progression through the grades is not automatic and some students repeat grades. Since our study covers both UBS and SSS we had students who were below and above the age range of the GYTS. Using the GYTS classification would mean excluding a huge number of our sample in the study, therefore it was not feasible to restrict our age classification to that of the GYTS.

Please revise the entire section on "Factors associated with exposure to SHS" to ensure that it reflects an associational study. Also, in this revision, you should not only indicate whether the relationship is significant or not but also you should include the direction of the association. E.g., X is significantly associated with increased/decreased Y.

This section has now been revised and read as follows:

Factors associated with SHS exposure at home and outside the home.

Factors associated with SHS exposure at home

As shown in Table 3, after adjusting for age, gender and school location, girls (OR 1.34, 95% CI 1.22 -1.47), students aged 18-20 years (OR 1.20, 95% CI 1.02 -1.40), those in UBS schools (OR 1.40, 95% CI 1.25 -1.57) and students attending grant –aided schools (OR 1.36, 95% CI 1.17 -1.58) were significantly more likely to be exposed to SHS. Living with parents (OR 0.83, 95% CI 0.74 -0.93), being a smoker (OR 1.63, 95% CI 1.31 -2.03), having smoking allowed at home and having family members or friends who smoked also significantly increased the risk of students exposure to SHS at home. In addition students who were sent to purchase cigarettes (OR 1.98, 95% CI 1.80-2.18) and supported a ban on public smoking were significantly more likely to be exposed to SHS at home.

Factors associated with SHS exposure outside the home

Outside the home, lower maternal and higher paternal educational level, living with parents and being sent to purchase cigarettes for others (OR 1.42, 95% CI 1.14 -1.77) were significantly associated with increased risk of exposure to SHS (Table 3). Additionally, older students aged 18-20 (OR 1.14, 95% CI 0.83 -1.56) were more likely to be exposed to SHS outside the home compared to younger students aged 12-14.

Discussion

1. The reference #17 on page 14 of 19 is likely wrong. Please look for correct and more appropriate reference.

Thank you for spotting out this mistake. We have updated the text as shown above and the new added reference is:

6.WHO Report on the Global Tobacco Epidemic: Monitoring tobacco use and prevention policies. 2017.(accessed December 2017)

2. Page 15 and 19, para 2: The authors should reconsider the suggestion of having government to develop policies for private homes as it could raise civil liberty and social issues.

This sentence has been deleted.

References

Although I will defer to the authors on this, references 17 and 18 utilize information from Saudi Arabia study, which may not be correct. As such, I will urge the authors to ensure that the references that they have used correctly support the information provided.

This was a mistake and reference 17 has been deleted and 18 has been moved to the right place.

Two additional references have been added, please see below new references:

6.WHO Report on the Global Tobacco Epidemic: Monitoring tobacco use and prevention policies. 2017.(accessed December 2017)

17.Tumwine J. Implementation of the framework convention on tobacco control in Africa: current status of legislation. Int J Environ Res Public Health. 2011;8(11):4312-31.

Minor comments

1. The usage of “determinants” in the title suggests some form of causation. The study by the authors can establish only some form of association. As such, the authors should revise the title to reflect the kind of study that they conducted.

Thank you for your suggestion. We have changed the title of the manuscript to “Prevalence and factors associated with exposure to Second-Hand Smoke (SHS) among young people: a cross-sectional study from the Gambia”.

2. Ensure consistency in the spelling of second-hand smoke (see page 8 of 19, ln 49).

This has now been revised.

3. Page 11 of 19, ln 34: Provide space between “faiths” and “(OR....)”

This has been revised.

4. Ensure consistency in the use of decimal places (see page 13 of 19, ln 52)

This has been changed and all figures have been rounded to one decimal place.

5. Page 15 of 19: “effect” on ln 42 should be “effective”.

This has been revised and the sentence now reads “Educating parents about the harmful effects smoking and exposure to SHS could be one of the effective ways to protect young people at home”.

6. Spell out all abbreviations in the tables and figure or include a note.

Footnotes have been added to all the tables and the figure.

7. Add “(N=10,289)” in the title of all the tables.

This has been added to all tables.

VERSION 2 – REVIEW

REVIEWER	Ellis Owusu-Dabo School of Public Health, Kwame Nkrumah University of Science and Technology Kumasi, Ghana
REVIEW RETURNED	02-Jan-2018

GENERAL COMMENTS	Apart from a few typos cited in the manuscript, previous concerns have been addressed.
--

REVIEWER	Hadii Mamudu East Tennessee State University, USA
REVIEW RETURNED	09-Jan-2018

GENERAL COMMENTS	1. The authors should replace words such as “many”, “majority” etc. with actual numbers. 2. The authors should ensure consistency in terms of font size, punctuation, capitalization, spacing etc. This will require the thorough line-by-line editing of the manuscript.
--

VERSION 2 – AUTHOR RESPONSE

Reviewer: 1

Reviewer Name: Ellis Owusu-Dabo

Institution and Country: School of Public Health, Kwame Nkrumah University of Science and Technology, Kumasi, Ghana

Competing Interests: None declared

Apart from a few typos cited in the manuscript, previous concerns have been addressed.

The suggested correction has been made.

Reviewer: 2

Reviewer Name: Hadii Mamudu

Institution and Country: East Tennessee State University, USA

Competing Interests: None declared

1. The authors should replace words such as "many", "majority" etc. with actual numbers.

These words has been replace with actual numbers in all applicable areas, which is highlighted in the revised mark manuscript.

2. The authors should ensure consistency in terms of font size, punctuation, capitalization, spacing etc. This will require the thorough line-by-line editing of the manuscript.

The authors has edited the manuscript line-by-line which is also highlighted in the revised mark manuscript.